# CMOS Depth Image Sensor with Offset Pixel Aperture Using a Back-Side Illumination Structure for Improving Disparity

**DOI:** 10.3390/s20185138

**Published:** 2020-09-09

**Authors:** Jimin Lee, Sang-Hwan Kim, Hyeunwoo Kwen, Juneyoung Jang, Seunghyuk Chang, JongHo Park, Sang-Jin Lee, Jang-Kyoo Shin

**Affiliations:** 1School of Electronics Engineering, Kyungpook National University, 80 Deahak-ro, Buk-gu, Daegu 41566, Korea; jmLee@ee.knu.ac.kr (J.L.); shkim7@knu.ac.kr (S.-H.K.); hwkwen@knu.ac.kr (H.K.); jyjang@knu.ac.kr (J.J.); 2Center for Integrated Smart Sensors, KAIST, 291 Daehak-ro, Yuseong-gu, Daejeon 34141, Korea; schang71@kaist.ac.kr (S.C.); parkjh20@kaist.ac.kr (J.P.); sjlee82@kaist.ac.kr (S.-J.L.)

**Keywords:** offset pixel aperture, CMOS depth image sensor, back-side illumination structure, improving disparity

## Abstract

This paper presents a CMOS depth image sensor with offset pixel aperture (OPA) using a back-side illumination structure to improve disparity. The OPA method is an efficient way to obtain depth information with a single image sensor without additional external factors. Two types of apertures (i.e., left-OPA (LOPA) and right-OPA (ROPA)) are applied to pixels. The depth information is obtained from the disparity caused by the phase difference between the LOPA and ROPA images. In a CMOS depth image sensor with OPA, disparity is important information. Improving disparity is an easy way of improving the performance of the CMOS depth image sensor with OPA. Disparity is affected by pixel height. Therefore, this paper compared two CMOS depth image sensors with OPA using front-side illumination (FSI) and back-side illumination (BSI) structures. As FSI and BSI chips are fabricated via different processes, two similar chips were used for measurement by calculating the ratio of the OPA offset to pixel size. Both chips were evaluated for chief ray angle (CRA) and disparity in the same measurement environment. Experimental results were then compared and analyzed for the two CMOS depth image sensors with OPA.

## 1. Introduction

Recent studies on image sensors have focused on depth cameras. Since the development of cameras, there has been consistent and continuous research on the advancement of imaging technology for capturing two-dimensional (2D) images, higher-dimensional images, etc. However, despite the improved quality of 2D imaging technology, depth information is not captured. Due to the inability of conventional 2D image sensors to capture the depth information of images, the quality of 2D images is lower compared with actual images seen by the human eye. Three-dimensional (3D) imaging technology complements the 2D image technology. 3D imaging technology can provide more stereoscopic image information to observers through depth information, which is not present in 2D images. Presently, a wide variety of studies are being conducted on 3D image sensor technology. Typical 3D imaging methods include time of flight (ToF), structured light, and stereo vision [1,2,3,4,5,6,7,8,9,10,11,12,13,14,15,16,17].

The ToF method is used to calculate the return time of incident light on an object and determine the distance using a high-power infrared (IR) source. Structured light imaging is a method of illuminating an object with a specific light pattern and analyzing it algorithmically to determine the object distance. Finally, a stereo camera is a setup that uses a plurality of cameras to obtain an image according to the incidence angle, which is used to obtain the distance. The above-mentioned methods are used as external elements to extract depth information to realize a 3D image. The manufacturing cost and the power consumption of 3D image sensors using these methods are relatively high because two or more cameras, structured light sources, or IR sources are required. However, depth image sensors adopting the offset pixel aperture (OPA) technique can extract depth information with a simple structure and without external elements. Previously, a sensor fabricated using the OPA technique was reported with a front-side illumination (FSI) structure [18,19,20]. Generally, in depth image sensing, disparity tends to be inversely proportional to pixel height. However, CMOS depth image sensors with OPA, which have the FSI structure, are limited in that the pixel height is reduced due to multiple metal layers on the front-side of the pixel. Therefore, a back-side illumination (BSI) structure is applied to OPA CMOS depth image sensors to improve disparity. The OPA CMOS depth image sensor with the BSI structure has a pixel height that is relatively lower than that of one with the FSI structure. If the pixel height is low, the difference in the peak angle of response between left-OPA (LOPA) and right-OPA (ROPA) and the disparity increases.

The rest of this paper is organized as follows. Disparity is described in Section 2. The design of the pixel structures is described in Section 3. Measurement of disparity based on distance and measurement of the chief ray angle (CRA) with OPA-based depth image sensors is described in Section 4. Finally, Section 5 concludes the paper.

## 2. Disparity

Depth information is the most important factor for realizing 3D images. A universalized 2D image sensor uses only the *x*- and *y*-axis information of the object to generate data. Therefore, the image data from the 2D camera is formed without distance information to the actual object information. To implement this in the 3D method, depth information must be applied to the planar image data. The depth information of the image is important data to detect the distance of the object. Therefore, when the depth information is extracted and applied to the image, it is possible to know the distance of the object. The OPA depth image sensor described in this paper used disparity to extract the depth information. In the OPA method, an aperture is applied to the inside of the pixel to obtain disparity information. The aperture integrated inside the pixel exhibits equivalent optical characteristics with the camera lens. Figure 1 shows that the effects of the OPA and lens aperture of a camera are equivalent. O_1_ is the offset value of the camera lens aperture and O_2_ is the offset value of the OPA.

Figure 2 shows a cross-sectional view of the pixel of the CMOS depth image sensor with OPA. The equivalent offset f-number (F_OA_) is inversely proportional to the pixel height and is given by:(1)$$F_{\mathrm{OA}}\;=\;\frac{f}{O_{\mathrm{CA}}}\;=\;\frac{h}{O_{p}}\;=\;\frac{h}{2O_{2}n},\;$$
where *f* is the focal length; O_CA_ is the same as 2 × O_1_; *h* is the pixel height; and O_P_ is an offset length between LOPA and ROPA. *n* is the refractive index of the microlens and planarization layer. Since O_CA_ is proportional to O_P_ divided by the pixel height, F_OA_ is proportional to the pixel height divided by O_P_. If the F_OA_ value is less than the f-number of the camera lens, light is not transmitted accurately to the OPA, and thus the disparity cannot be correctly obtained. Therefore, F_OA_ is set to a number greater than the f-number of the camera lens.

Figure 3 shows the effect of object distance and image distance on disparity. *a* is the object distance in a non-focusing state, and *a*_0_ is the object distance in a focusing state. *b* is the image distance in a non-focusing state, and *b*_0_ is the image distance in a focusing state. *d* is the distance between a sensor and the focus in a non-focusing state. Definitions of disparity are as follows:(2)$$\frac{1}{a}\;+\;\frac{1}{b}\;=\;\frac{1}{f}\;\to \;b\;=\;\frac{af}{a-\;f},\;$$
(3)$$\frac{1}{a_{0}}\;+\;\frac{1}{b_{0}}\;=\;\frac{1}{f}\;\to \;b_{0}\;=\;\frac{a_{0}f}{a_{0}-\;f},$$
(4)$$d_{dis}=\;\frac{d}{b}O_{CA}\;=\;\frac{f^{2}\left(a-a_{0}\right)}{a\left(a_{0}-f\right)}\;\times \;\frac{1}{F_{OA}},\;$$
where *d_dis_* is the disparity, which can be calculated from the lens equation. In particular, as the pixel height decreases, F_OA_ decreases, and thus the disparity increases.

## 3. Design of the Pixel Structure

The disparity varies depending on the size of the F_OA_, as expressed in the equation described in the previous section. Disparity increases when the pixel height decreases or the offset value of the OPA increases. Therefore, when designing a pixel using the OPA method, it is essential to accurately set the pixel height and the offset value of OPA. Two different sensors were used for measurement.

Figure 4a,b show a cross-sectional illustration of pixel height differences depending on the fabrication process. As shown in Figure 4a, the pixel height of the CMOS depth image sensor with OPA using the FSI structure used for measurement in this work was 4.435 μm. The CMOS depth image sensor with OPA using the FSI fabrication process has multiple metal layers for signal transmission and power voltage on the front side of the pixel.

These metal layers are an essential element for driving the sensor. Therefore, the decrease in pixel height for this type of sensor is limited. As above-mentioned, to increase disparity, the offset value of the OPA must be increased or the pixel height must be reduced. The increasing offset value of OPA is limited. It is possible to increase the offset value of OPA within a pixel area. If the offset value of OPA is increased, the opening region of the OPA is reduced accordingly, and there is a risk that sensitivity will decrease. Therefore, the method of decreasing the pixel height has advantageous characteristics without the risk of the CMOS depth image sensor with OPA, as described previously. If the BSI structure is applied to the CMOS depth image sensor with OPA, light is transferred through the microlens to the back-side of the pixel, even though the multiple metal layers are applied to the front-side of the pixel. Therefore, the pixel height can be considerably reduced. As shown in Figure 4b, the pixel height of the CMOS depth image sensor with OPA using the BSI structure used in this measurement was 1.16 μm. A drawback of the BSI structure applied to reduce the pixel height is the crosstalk of light passing through the color filters. In particular, there is a possibility that a light component that has passed through the color filter may flow into the photodiode of an adjacent pixel, resulting in a decrease in color response. This can be complemented by forming a deep trench isolation (DTI) region between adjacent photodiodes.

## 4. Measurement Results and Discussion

This section describes the results of the measurements using two chips with different structures. The color patterns of the two chips form a RGBW pattern, as shown in Figure 5. If the LOPA and ROPA channels are separated to calculate the disparity, the resolution of the OPA pixel images will be lower than the resolution of the entire image. The entire pixel area can be composed of only LOPA and ROPA patterns, but the sensitivity of the image sensor is reduced. Therefore, for each 2 × 2 area, one OPA pixel is integrated into the OPA image sensor with a color pattern. In other words, 2D quality improves as the density of OPA pixels decreases, but there is a trade-off in which the disparity resolution and depth accuracy decrease. Two types of chips used in the measurement were manufactured via different fabrication processes. Therefore, the pixel pitch is different. The pixel pitch of the CMOS depth image sensor with OPA using the FSI structure was 2.8 μm × 2.8 μm. The offset value of the OPA was 0.65 μm, and the OPA opening width was 0.5 μm. To compare the two chips, a BSI chip with similar conditions must be selected. The ratio of the OPA offset to pixel area is 46.42% in the CMOS depth image sensor with OPA using the FSI structure. The pixel pitch of the CMOS depth image sensor with OPA using the BSI structure was 1 μm × 1 μm. For the CMOS depth image sensor with OPA using the BSI structure, a similar offset ratio value to FSI structure was 0.24 μm. The ratio of OPA offset to pixel area was 48% and the OPA opening width was 0.26 μm.

CRA and disparity measurements were performed using the above-mentioned chips. In this measurement, CRA is the ray angle that passes through the center of an OPA. The response of the sensor can be estimated by the incidence angle from the light source, as shown in Figure 6. Therefore, the angle response of LOPA and ROPA can be confirmed using CRA measurements. Figure 6 shows the measurement environment of the CRA. A connected collimator was designed to transmit light in parallel to the sensor. Angle of incident light is an important factor in the CRA measurement environment. Therefore, it is essential to align the center of the collimator and the center of the sensor vertically and horizontally. As shown in Figure 6, the angle of the sensor is controlled after aligning the center of the sensor to the center of the collimator. This is the same effect on changing the angle of incident light in the measurement environment. The CRA measurement range was −30° to 30°, and the measurement was performed at 2° intervals.

Figure 7 shows the CRA measurement results of the CMOS depth image sensor with OPA using the FSI structure. For LOPA, maximum output was obtained when the incidence angle was −24°. The maximum output of ROPA was obtained when the incidence angle was 20°. Therefore, the peak angle response difference between the LOPA and ROPA in the CMOS depth image sensor with OPA using the FSI structure was 44°. In this result, the origin was shifted because the position of the microlens was also shifted due to the error in the fabrication process.

Figure 8 shows the CRA measurement results of the CMOS depth image sensor with OPA using the BSI structure. For LOPA, maximum output was obtained when the incidence angle was −26°. The maximum output of ROPA was obtained when the incidence angle was 26°. Therefore, the peak angle response difference between the LOPA and ROPA in the CMOS depth image sensor with OPA using the BSI structure was 52°. Following the results of the CRA measurement, the peak angle response of the CMOS depth image sensor with OPA using the BSI structure was 8° higher than that with the FSI structure.

Figure 9 shows a cross-sectional view of the correlation between pixel height, offset value of OPA, and response angle. The formula for the response angle (*α*) is given below:(5)$$\alpha =\;n\frac{O_{2}}{h},$$
where *h* is the pixel height; O_2_ is the offset value of OPA; and *n* is the refractive index of the microlens and planarization layer or SiO_2_ layer. The response angle of the OPA pixel as expressed in the formula can be modified by adjusting the offset value of OPA or the pixel height. Therefore, the range of the CMOS depth image sensor’s peak-to-peak response angle with the BSI structure is wider than that of the CMOS depth image sensor with OPA using the FSI structure.

Disparity is the most important factor in obtaining depth information with the CMOS depth image sensor with OPA to realize the 3D image. Improving disparity also improves the depth of information resolution. Disparity measurements were performed to confirm that the disparity of the CMOS depth image sensor with OPA using the BSI structure that had a low pixel height was improved compared to the CMOS depth image sensor with OPA using the FSI structure. Figure 10 shows the measurement environment for disparity evaluation; the object is a black-and-white pattern for the disparity measurement because the monotonic boundary area makes it easy to measure disparity. The measurement environments for both sensors are equally configured. First, the camera lens of the CMOS depth image sensors with OPA (using the FSI and BSI structures, respectively) is placed 100-cm away from the object and focused. Then, the disparity according to the distance is measured from 86 to 114 cm at 1-cm intervals using a step motor rail. It is possible to extract the disparity over a range of 14 cm from the focal length, but if it is too far from the focal point, blurring of the image becomes significant and it is difficult to calculate the accurate disparity value.

Figure 11 shows the disparity measurement results for the OPA depth image sensors with the FSI structure and the BSI structure, respectively. Based on the position of 100 cm, which is the focal point on the *x*-axis of the graph, it is shown from −14 cm to 14 cm at 1 cm intervals. The disparity was obtained by separating the LOPA and ROPA images from the entire images and calculating the difference between the images in the boundary area of the object. The disparity increases because of the blurring of images due to the distance of the sensors from the focal point. Therefore, the disparity at the farthest point from the focal point was compared in the measurement result. For the CMOS depth image sensor with OPA using the BSI structure, the disparity was 6.91 pixels when the distance between the sensor and the focal point was −14 cm. When the distance between the sensor and the focal point was 14 cm, the disparity was 3.53 pixels. The disparity for the CMOS depth image sensor with OPA using the FSI structure was 1.73 pixels while the distance between the sensor and the focal point was −14 cm. When the distance between the sensor and the focal point was 14 cm, the disparity was 0.89 pixels. Accordingly, the disparity of the CMOS depth image sensor with OPA using the BSI structure increased by 5.18 pixels at −14 cm and 2.63 pixels at 14 cm from the focal point.

Following the measurement results of the disparity, the disparity of the CMOS depth image sensor with OPA using the BSI structure, which has a low pixel height, was increased. As can be seen from Equations (1) and (4), as the pixel height decreases, the F_OA_ value decreases, and the disparity value increases.

The disparity obtained as described above can be converted into distance information. First, it is important to derive the effective F_OA_ value to convert it into distance information. The formula for the F_OA_ is given below:(6)$$F_{OA}=\;\frac{f^{2}\left(a-a_{0}\right)}{a\left(a_{0}-f\right)\times \left[\frac{d_{dis}\times \;d_{pixel}}{1000}\right]},\;$$
where *f* is the focal length; *a* is the object distance in a non-focusing state; and *a*_0_ is the object distance in a focused state. *d_dis_* is the disparity, and *d_pixel_* is the distance between the LOPA and ROPA images, depending on the pixel pitch. The OPA CMOS depth image sensor with the BSI structure had an effective F_OA_ value of 5.7 and the OPA CMOS depth image sensor with FSI structure had an effective F_OA_ value of 6.9. At present, it is difficult to directly compare the disparity of the BSI structure with that of the FSI structure because the fabrication process and pixel pitch of two structure are different. Therefore, the F_OA_ value of the OPA CMOS depth image sensor with the FSI structure was adjusted to provide the conditions—excluding the pixel height of the FSI structure—that are consistent with the BSI structure. The adjusted F_OA_ value of the OPA CMOS depth image sensor with the FSI structure was 18.63. These set conditions were calculated and compared with the measured distance values for the FSI structure and BSI structure. The formula for the object distance is given below:(7)$$a=\;\frac{f^{2}a_{0}}{f^{2}-F_{OA}\left(a_{0}-f\right)\times \left[\frac{d_{dis}\times \;d_{pixel}}{1000}\right]}$$

The results of the measured distance using the equation above and the measured disparity is shown in Figure 12. When the actual distance was 86 cm, the measured distance with the BSI structure was 87.40 cm and the measured distance with the FSI structure was 89.28 cm. When the actual distance was 114 cm, the measured distance with the BSI structure was 106.18 cm and the measured distance with the FSI structure was 104.88 cm. The results revealed that the measured distance result of the OPA CMOS depth image sensor with BSI structure was further improved.

A comparison with the disparity tendencies calculated under similar conditions with a different focal point was also performed for a clearer analysis. Figure 13 shows the results of the calculated disparity with different focal points using effective F_OA_ values. The focal point was 40 cm and the calculation range was from 40 to 80 cm. The calculated results reveal that the disparity of the CMOS depth image sensor with OPA using the BSI structure increased by 22.97 pixels at 80 cm.

Disparity resolution is an important characteristic change due to the decreasing pixel height by applying the BSI structure. Figure 14 shows the results of the disparity resolution between the FSI and BSI structures due to the difference in pixel height under the same conditions. For the calculation conditions, the focal point was 40 cm, and the disparity range was from −5 to 5 pixels at 1-pixel intervals. The disparity resolution results revealed that when the disparity increased by one pixel, the disparity resolution of the BSI sensor was 0.43 cm and the disparity resolution of the FSI sensor was 1.42 cm. However, the disparity resolution of the BSI sensor had been improved compared to the FSI sensor. The summary of two chips with different structures is given in Table 1.

## 5. Conclusions

In this study, we examined two OPA-based CMOS depth image sensors with different side illumination structures to improve disparity. The CMOS depth image sensors with OPA for the measurement were designed and manufactured via different fabrication processes. The previously reported CMOS depth image sensor with OPA was designed via the FSI fabrication process. The difference in pixel height caused by the presence or absence of a color filter was compared in the FSI structure. As the pixel height of the OPA sensor without the color filter was lower than that of the OPA sensor with the color filter, the CRA increased. However, the decrease in pixel height is limited depending on the structural characteristics of FSI. The pixel height of the CMOS depth image sensor with OPA using the BSI structure can be reduced without affecting the multiple metal layers. The pixel height of the CMOS depth image sensor with OPA using the BSI structure used for measurement in this work was lower than that of the one with the FSI structure by 3.035 μm. The color filter patterns of the FSI chip and BSI chip manufacturing processes were the same as the RGBW pattern for improved sensitivity. The two chips used for measurement did not have the same pixel size and fabrication process, so it is difficult to make a quantitative comparison. Due to the multiple metal layers and high pixel height, the signal-to-noise ratio (SNR) of the CMOS image sensor with OPA using FSI structure was inferior to that of the BSI structure under the same conditions. If the SNR of the image is not good, the disparity between LOPA and ROPA cannot be acquired accurately. Therefore, the disparity resolution is expected to be improved with better SNR.

The results of the measurements show that the CRA and disparity values increase as the pixel height decreases. In particular, the results of the CRA measurement show that the peak angle response of the CMOS depth image sensor with OPA using the BSI structure is 52°, and the peak angle response of the CMOS depth image sensor with OPA using the FSI structure is 44°. Accordingly, the peak angle response value of the sensor with the BSI structure was extended by 8° because the incidence angle of the microlens increased as the pixel height decreased. The disparity of the CMOS depth image sensor with OPA using the BSI structure was greater than that of the CMOS depth image sensor with OPA using the FSI structure by 5.18 pixels at −14 cm and 2.63 pixels at 14 cm from the focal point. This is because the pixel height and the F_OA_ values of the sensor with the BSI structure are decreased. Assuming all other conditions except the pixel height are the same, disparity resolution increases as F_OA_ decreases. Therefore, high-quality depth information can be obtained with a single sensor by applying the BSI structure to the CMOS depth image sensor with OPA. In conclusion, the CMOS depth image sensor with OPA using the BSI structure can be easily applied to portable devices or other fields to improve depth information quality as well as to achieve low fabrication costs and low power consumption with a simple structure and without external elements.

In future experiments, we plan to analyze the characteristics of various color patterns such as RBWW and monochrome patterns as well as RGBW patterns.

## Figures and Tables

**Figure 1 sensors-20-05138-f001:**
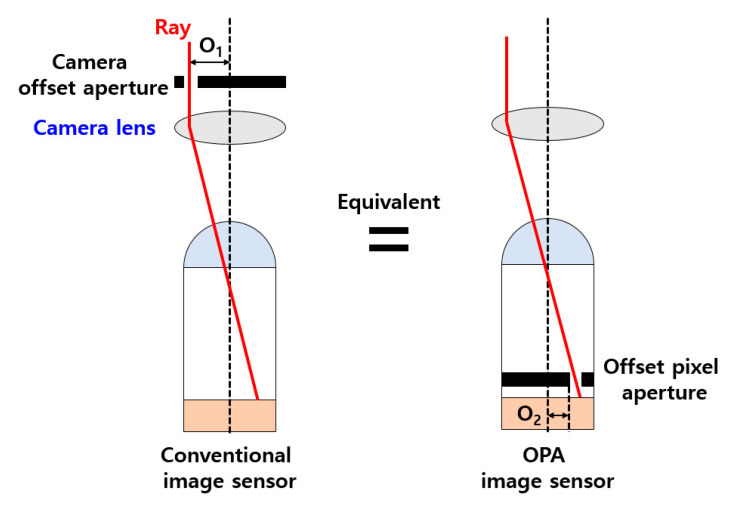
Equivalent effect between OPA and camera lens aperture.

**Figure 2 sensors-20-05138-f002:**
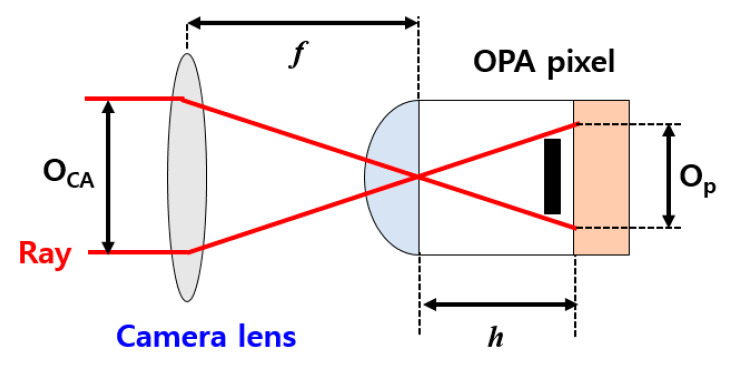
Cross-sectional view of the pixel of the CMOS depth image sensor with OPA.

**Figure 3 sensors-20-05138-f003:**
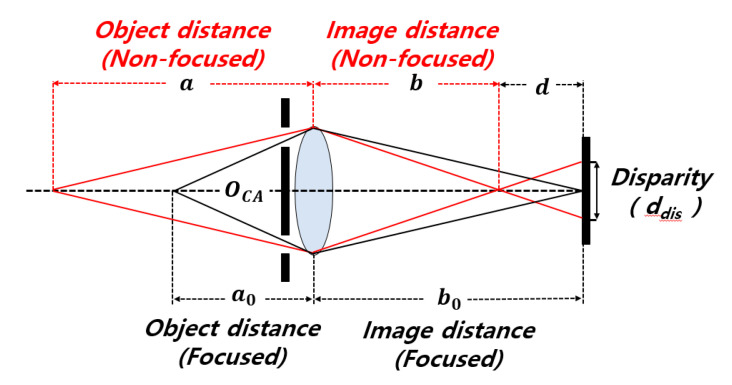
Effect of object distance and image distance on disparity.

**Figure 4 sensors-20-05138-f004:**
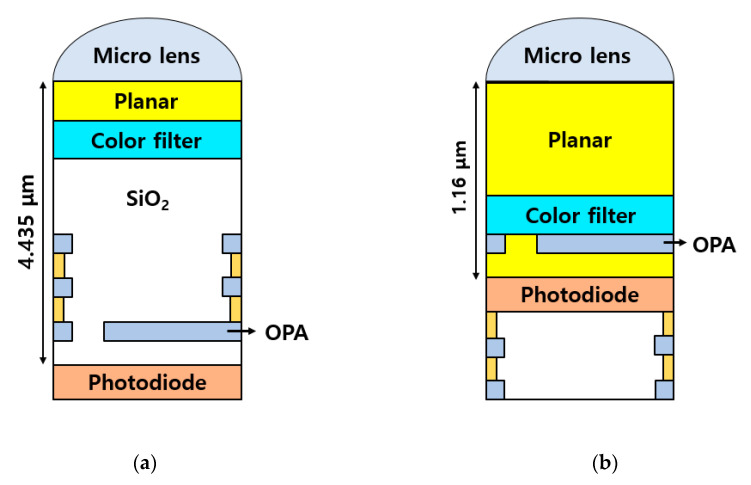
Cross-sectional illustration of pixel height differences depending on the fabrication process. (**a**) CMOS depth image sensor with OPA using an FSI structure. (**b**) CMOS depth image sensor with OPA using a BSI structure.

**Figure 5 sensors-20-05138-f005:**
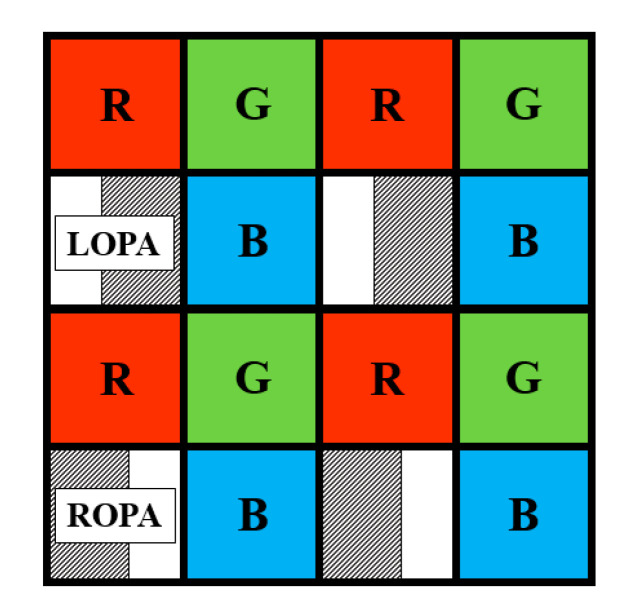
Color pattern of two chips with different structures.

**Figure 6 sensors-20-05138-f006:**
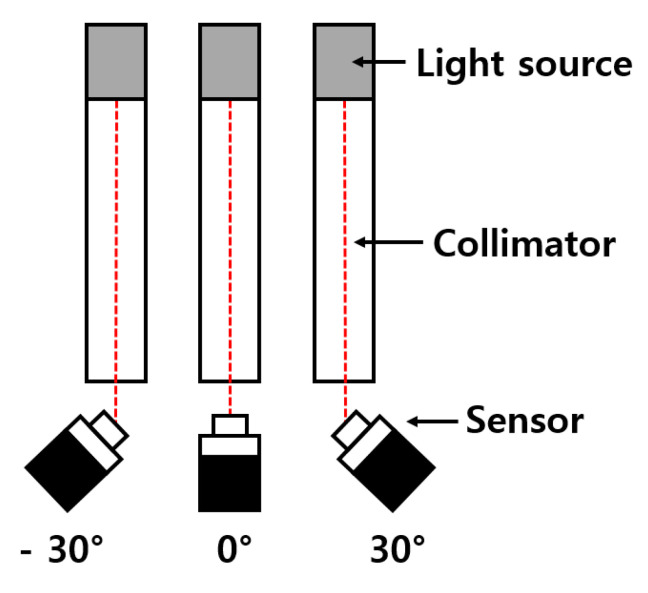
Measurement environment of the CRA.

**Figure 7 sensors-20-05138-f007:**
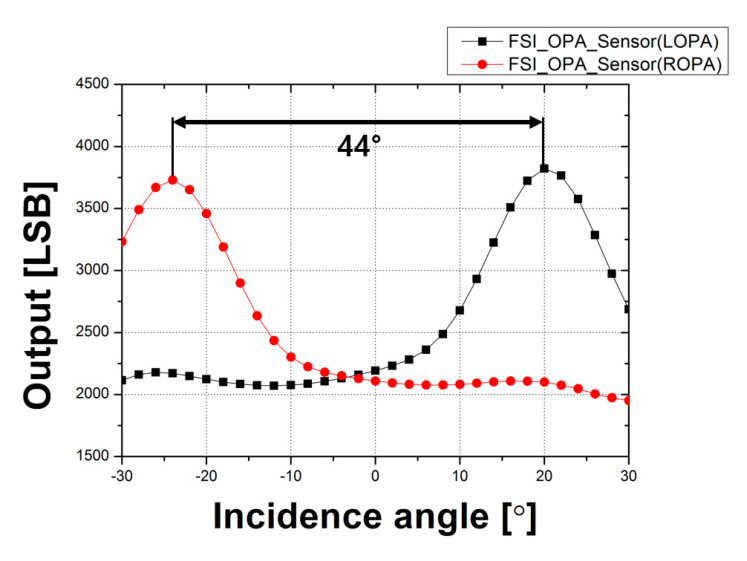
CRA measurement results of the CMOS depth image sensor with OPA using the FSI structure.

**Figure 8 sensors-20-05138-f008:**
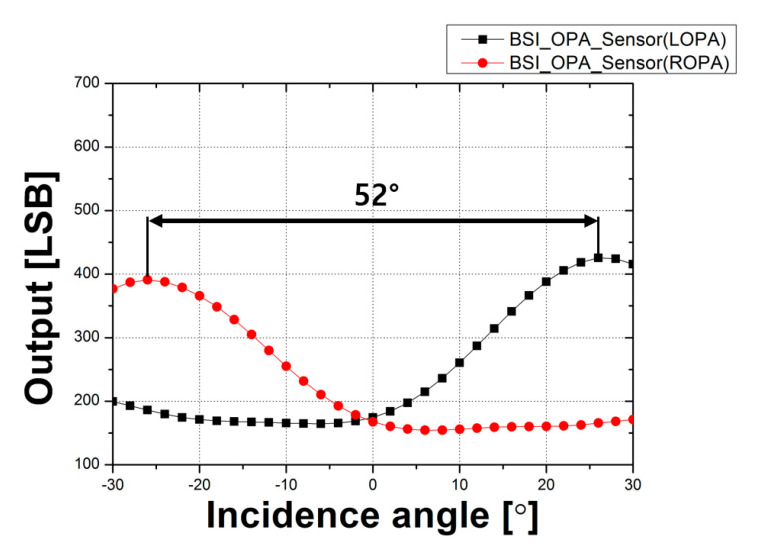
CRA measurement results of the CMOS depth image sensor with OPA using the BSI structure.

**Figure 9 sensors-20-05138-f009:**
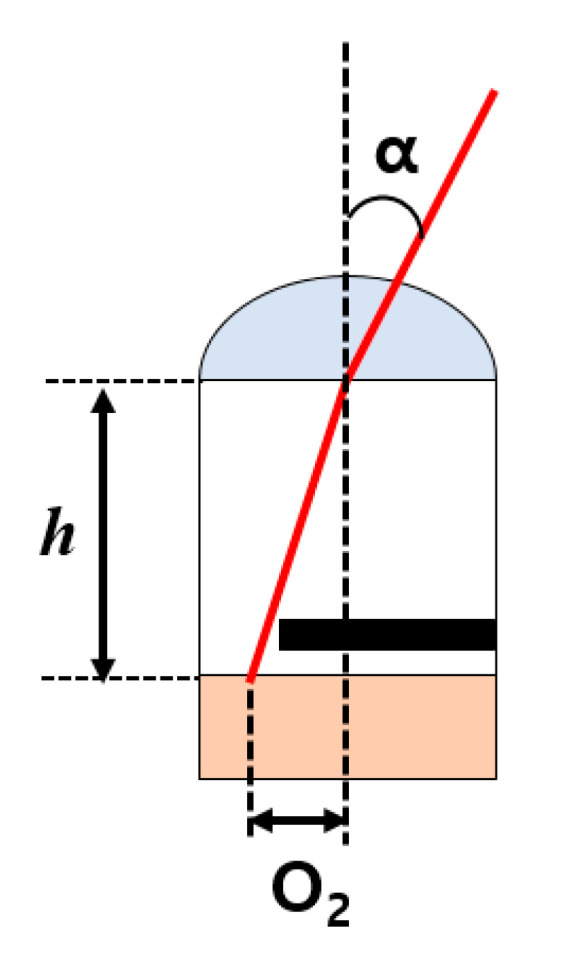
Cross-sectional view of the correlation between pixel height, offset value of OPA, and response angle.

**Figure 10 sensors-20-05138-f010:**
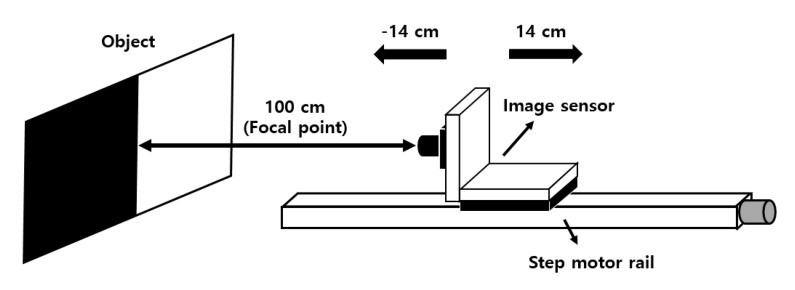
Measurement environment for disparity evaluation.

**Figure 11 sensors-20-05138-f011:**
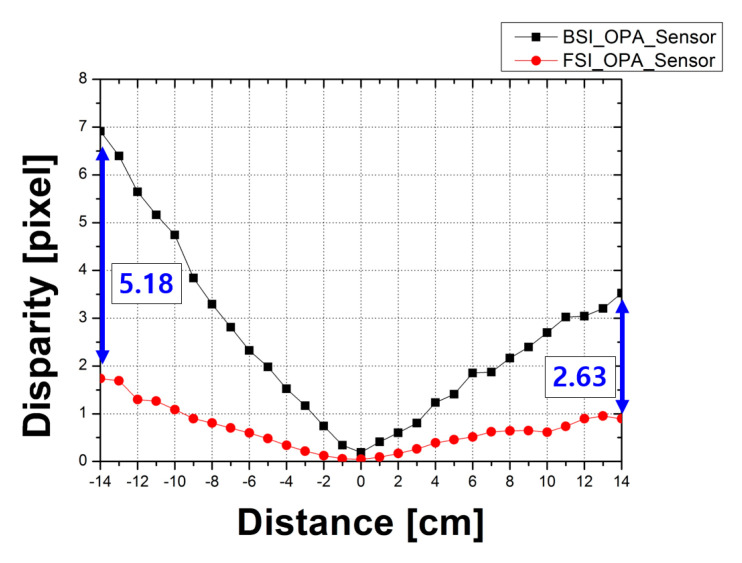
Disparity measurement results of the CMOS depth image sensor with OPA using the FSI structure and the BSI structure.

**Figure 12 sensors-20-05138-f012:**
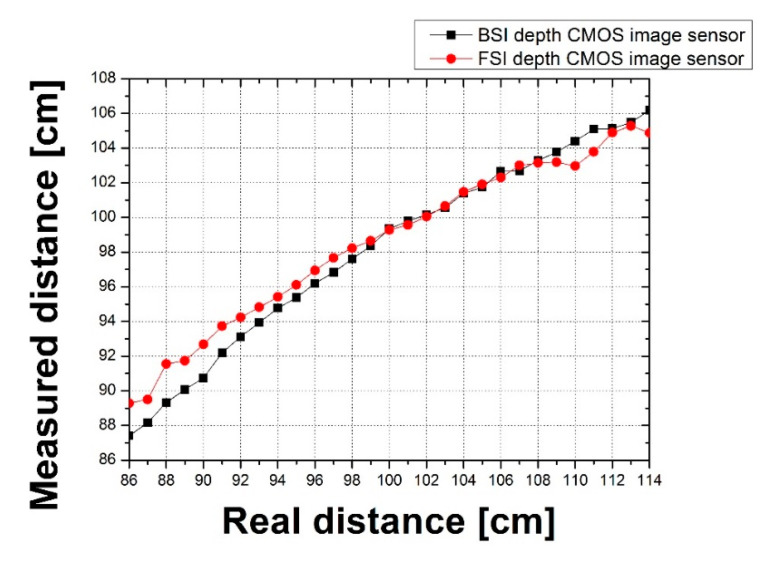
Results of the measured distance calculated from the measured disparity.

**Figure 13 sensors-20-05138-f013:**
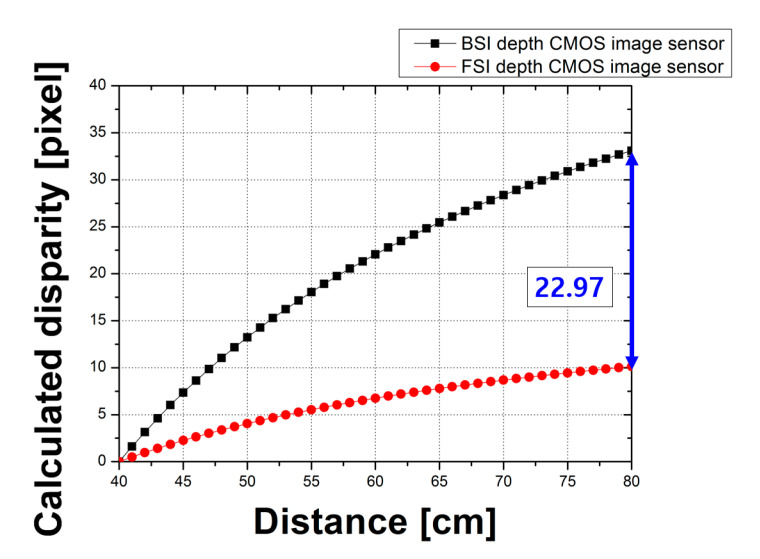
Results of the calculated disparity with different focal points using effective F_OA_ values.

**Figure 14 sensors-20-05138-f014:**
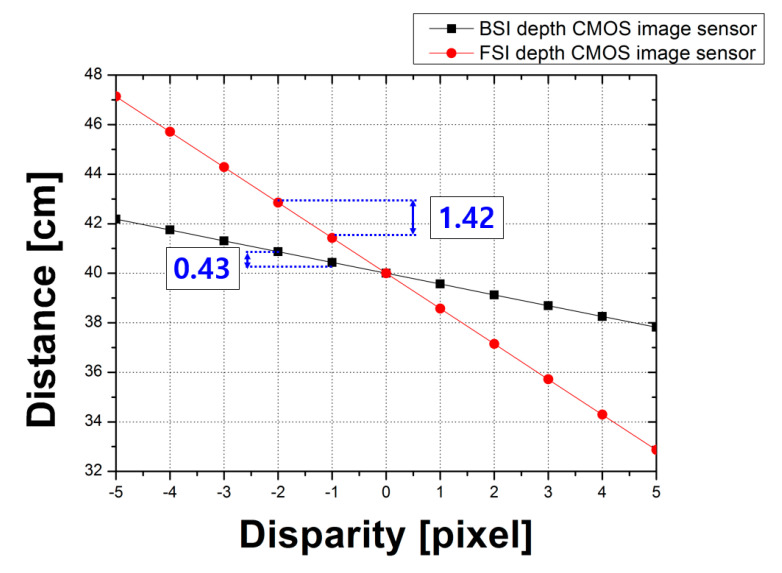
Results of the disparity resolution between the FSI and BSI structures due to the difference in pixel height under the same conditions.

**Table 1 sensors-20-05138-t001:** Summary of the two chips with different structures.

	Front-Side Illumination Structure	Back-Side Illumination Structure
Pixel size	2.8 μm × 2.8 μm	1 μm × 1 μm
Fabrication process	FSI process	BSI process
Pixel resolution	1632 (H) × 1124 (V)	4656 (H) × 3504 (V)
Color pattern	RGBW pattern	
Pixel height	4.435 μm	1.16 μm
Effective equivalent offset f-number	6.9	5.7
Peak response angle	44°	52°
Disparity (−14 cm) *	1.73 pixels	6.91 pixels
Disparity (14 cm) *	0.89 pixels	3.53 pixels

* This number denotes the sensor position from the focal point.

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
