# Peer review of "CMOS Depth Image Sensor with Offset Pixel Aperture Using a Back-Side Illumination Structure for Improving Disparity"

_sensors, 2020, doi:10.3390/s20185138_

Round 1

Reviewer 1 Report

In this paper, authors present a CMOS depth image sensor with offset pixel aperture (OPA) using a back-side illumination structure for improving disparity. In general terms, this work is very interesting, it is well written, structured and it is clear about the significance and its novelty; however, I strongly encourage authors to consider the following two recommendations:

1. The paper should include more information about its contribution and its relationship with the state-of-the-art methods.

2. In addition, I recommend to include a related work section.

Reviewer 2 Report

Please see my comments below. In particular some revision of the conclusions and the language seems to be beneficial for the paper before publishing.

  • more details on sensors that are characterized such as resolution, sensitivity (quantum efficiency) etc
  • point out some trade off between effective resolution and disparity resolution (references)
  • provide some explanations about the depth extraction
    • how is disparity converted into depth
    • the reader might expect a graph that shows distance versus distance for example
    • how does the sensor perform beyond the +/- 14cm distance range as shown in the paper with regard to disparity / depth extraction
  • when comparing FSI with BSI I am wondering about pixel SNR trade off versus disparity resolution
    • how is SNR related to depth resolution?
    • if not part of the study, please point to appropriate references
  • conclusion to be revised and supported by data
    • most part of the conclusion is a summary right now, please add conclusions on the impact for next investigations, challenges to solve. what are the drawbacks of the BSI sensor that have to be addressed in an improved version?
    • support conclusions on for example low power by providing data for the particular sensors that are compared
  • please revise the English language

Reviewer 3 Report

see attached file.

Round 2

Reviewer 3 Report

All the issues concerned are addressed appropriately. It is recommended to be published as it is.